# Sintering Process Optimization for 3YSZ Ceramic 3D-Printed Objects Manufactured by Stereolithography

**DOI:** 10.3390/nano11010192

**Published:** 2021-01-14

**Authors:** Sang Hyun Ji, Da Sol Kim, Min Soo Park, Ji Sun Yun

**Affiliations:** 1Energy & Environmental Division, Korea Institute of Ceramic Engineering and Technology, 101, Soho-ro, Jinju 52851, Korea; sanghyun_ji@kicet.re.kr (S.H.J.); maname89@naver.com (D.S.K.); 2Department of Mechanical System Design Engineering, Seoul National University of Science and Technology, Seoul 01811, Korea; pminsoo@seoultech.ac.kr

**Keywords:** 3D printing, sintering, 3YSZ, flexural strength, relative density

## Abstract

A 3YSZ (3 mol% yttria-stabilized zirconia) ceramic green body with 50 vol% of ceramic content was 3D-printed by supportless stereolithography under optimal drying, debinding, and sintering conditions in order to achieve high strength and density. The viscosity and flowability of the ceramic nanocomposite resins were optimized by adjusting the amounts of non-reactive diluents. The ceramic 3D-printed objects have a high polymer content compared to ceramics samples manufactured by conventional manufacturing processes, and the attraction between layers is weak because of the layer-by-layer additive method. This causes problems such as layer separation and cracking due to internal stress generated when materials such as solvents and polymers are separated from the objects during the drying and debinding processes; therefore, the drying and debinding conditions of 3YSZ ceramic 3D-printed objects were optimized based on thermogravimetry–differential thermal analysis. The sintering conditions at various temperatures and times were analyzed using X-ray diffraction, SEM, and flexural strength analysis, and the body of the 3YSZ ceramic 3D-printed object that sintered at 1450 °C for 150 min had a relative density of 99.95% and flexural strength of 1008.5 MPa. This study widens the possibility of manufacturing ceramic 3D-printed objects with complex shapes, remarkable strength, and unique functionality, enabling their application in various industrial fields.

## 1. Introduction

Ceramic products are used in various fields such as electronics, aerospace, and biotechnology owing to their excellent strength and unique functionality [1,2,3]. In particular, ZrO_2_ ceramics have broad applications owing to their high strength, toughness, and chemical stability [4]. Engineering applications generally require ZrO_2_ ceramics with complex geometric shapes, but the high brittleness and hardness of ZrO_2_ ceramics make it difficult to produce complex-shaped parts by traditional methods [5]. Additive manufacturing (AM), also known as 3D printing, is an innovative method for industrial production as it can produce lighter and stronger parts and systems. AM can be an alternative to traditional manufacturing for obtaining a larger variety of ceramic products. There are various AM methods such as fused deposition modeling, selective laser sintering, and stereolithography apparatus (SLA) [6]. Among these, the SLA method involves a layer-by-layer buildup of a photopolymer resin in a desired shape by irradiation with a UV laser or a digital light processing projector. Compared to other AM methods, SLA is more commonly used in various fields such as medical implants and mechanical microparts, where high-precision parts are required [7] owing to its fast printing speed and high precision. Several studies on the SLA printing method have been reported for manufacturing high-strength ceramic 3D-printed objects by dispersing ceramic particles such as Al_2_O_3_ and ZrO_2_ in a photopolymer base resin [8,9]. The ceramic 3D printing method can produce complex and various shapes of ceramic parts that cannot be manufactured by conventional ceramic processes such as slip casting or tape casting. Furthermore, compared to 3D printing technology using only conventional photopolymer resins, ceramic 3D-printed objects can provide remarkable strength and unique ceramic functionality. These features facilitate the production 3D-printed objects with various functions.

To obtain a ZrO_2_ ceramic 3D printing sintered body with excellent physical properties using the SLA 3D printing method, several important factors need to be considered. First, the ZrO_2_ ceramic should be uniformly dispersed in the photopolymer base resin. The affinity between inorganic ceramic particles and organic photopolymer base resins is generally poor, and ceramic particles in ceramic composite resins tend to precipitate without being uniformly dispersed. The heterogeneous distribution of the ceramic particles in the composite resin negatively affects the strength of the 3D-printed objects because of non-uniformity between the additive layers. Second, the ceramic content in the ceramic composite resin should be optimized to fabricate high-density high-strength ceramic 3D-printed sintered objects. In most photopolymer composite resins for ceramic 3D printing, as the ceramic content increases, the 3D printing process becomes impractical due to poor dispersibility and high viscosity. Therefore, low-viscosity resins with a low ceramic content are mainly used. However, when a low-viscosity resin with a low ceramic content is utilized, the shape of the 3D-printed objects cannot be maintained after the sintering process. Hence, ceramic composite resins with a ceramic content of at least 40% are required for high strength and quality. Finally, it is very important to optimize the sintering process for ZrO_2_ ceramic 3D-printed objects because the physical properties of ceramic 3D-printed sintered objects are significantly influenced by the drying, debinding, and sintering process conditions. Compared to typical ceramic samples, the polymer content of ZrO_2_ 3D-printed objects is very high, and the layer-by-layer additive method results in weak binding forces between layers. Consequently, during the drying and debinding processes, internal stresses can occur when high-activity materials such as solvents, residues, and polymers quickly separate from the objects, which can frequently cause defects such as layer separation and cracking. Furthermore, the sintering temperature and sintering time should be optimized because they directly influence the formation of the grain and grain boundaries of the sintered body.

In this study, ceramic nanocomposite resins were manufactured using 3 mol% yttria-stabilized zirconia (3YSZ) with a ceramic content of 50 vol% based on 1,6-hexanediol diacrylate (HDDA) monomer, and the sintering process was optimized with the aim of fabricating high-strength and high-density SLA 3D-printed objects. According to the sintering temperature, ZrO_2_, a polymorphic material, undergoes phase changes from monoclinic (0–950 °C) to tetragonal (1200–2370 °C) and cubic (2370 °C melting point), resulting in a volume expansion of up to 4.5% [10]. Since the volume expansion that occurs during the sintering process generally causes ZrO_2_ objects to crack, stabilizers such as CaO, MgO, CeO_2_, and Y_2_O_3_ are added to stabilize the ZrO_2_ to prevent cracking and obtain excellent strength [11]. In particular, 3 mol% yttria-stabilized tetragonal ZrO_2_ has excellent durability and is widely used as a structural material or dental material [12]. In this regard, for the sintering stability of ZrO_2_, 3YSZ nanoparticles with 3 mol% yttria added as a stabilizer were used in this study. Dispersion stability and supportless 3D printing technology for high-viscosity ceramic nanocomposite resins with high ceramic content were optimized in our previous work [13,14]. By controlling the amount of non-reactive diluents, the viscosity and flowability of UV-curable 3YSZ ceramic nanocomposite resins were optimized for the supportless SLA 3D printing process. Furthermore, the drying and debinding conditions of 3YSZ ceramic 3D-printed objects were investigated based on thermogravimetry–differential thermal analysis (TG-DTA) to minimize the defects, and the optimization of sintering temperature and sintering time was carried out using X-ray diffraction (XRD), SEM, and flexural strength analysis to obtain good physical properties.

## 2. Material and Methods

Submicron tetragonal ZrO_2_ powder (crystal size 40 nm) stabilized with 3 mol% Y_2_O_3_ (3YSZ, Tosoh Corporation, Minato-Ku, Japan) was used as a ceramic nanopowder, and vinyltriethoxysilane (VTES) was used as the silane coupling agent for the surface modification of ceramic particles. HDDA (Sigma-Aldrich, St. Louis, MO, USA) was used as an acrylate-based monomer, and phenylbis(2,4,6-trimethylbenzoyl)phosphine oxide (BAPO, Sigma-Aldrich, St. Louis, MO, USA) was used as the photoinitiator. BYK-111 (BYK, Geretsried, Germany) and butoxyethanol (Bu, Sigma-Aldrich, St. Louis, MO, USA) were used as dispersants and non-reactive diluents, respectively.

3YSZ ceramic nanoparticles were surface-modified with acrylate groups via the hydrolysis and condensation of VTES [13,14]. For the hydrolysis and condensation reactions, mixtures of VTES, ethanol, and distilled water in 1:7.5:91.5 weight ratios were first vigorously stirred, and then 3YSZ ceramic nanoparticles were added to the VTES solution at 35 wt%. The mixture was then subjected to hydrothermal treatment at 100 °C for 3 h and dried under vacuum at 100 °C for 24 h. For fabrication of the UV-curable base resins for the SLA 3D printing process, HDDA monomer was vigorously mixed with the photoinitiator BAPO (added at 2 wt% of monomer), the dispersant BYK111 (added at 2 wt% of particles), and the non-reactive diluent Bu (added at 0–10 vol% of monomer). The VTES-coated 3YSZ nanoparticles (50 vol% of monomers) were then added to the base resins and uniformly mixed for 4 h at 400 rpm using a planetary mixer. The sample names of the UV-curable 3YSZ ceramic resins according to Bu contents were defined as shown in Table 1, and the sample names of ceramic resins were defined as 3YSZ-Bu*x*%, where 3YSZ, Bu, and x refer to 3Y-ZrO_2_ powder, Bu, and Bu contents, respectively.

UV-curable 3YSZ ceramic nanocomposite resins were exposed to a UV LED light with an intensity of approximately 12.7 mW at a wavelength of 405 nm for 1.3 s, and 3YSZ ceramic 3D-printed objects were fabricated in a layer-by-layer additive manner with layer thickness of 50 μm. The 3YSZ green body was washed with isopropyl alcohol and post-cured for 2 h. To minimize the internal stress generated from the removal of solvents and residues, the post-cured green body was dried under vacuum for 24 h at 25 °C. Further, to minimize the internal stress generated from the removal of organic materials, the dried green body was calcined by a three-step debinding process at 300 °C for 3 h, 380 °C for 3 h, and 700 °C for 1 h at a heating rate of 0.83 °C/min. After debinding, the 3YSZ 3D-printed objects were sintered at 700 °C for 1 h, 1000 °C for 1 h, and 1450 °C for 180 min for densification and grain growth.

The viscosity of the UV-curable 3YSZ ceramic nanocomposite resins was measured using a rheometer (MCR-502, Anton Paar, Graz, Austria) at shear rates of 10–50 s^−1^. Photopolymerization properties of the 3YSZ nanocomposite resins were observed by photo-DSC (photo-DSC 204 F1 Phoenix, NETZSCH, Selb, Germany) at a wavelength of 355 nm. The flowability of the 3YSZ nanocomposite resins was measured as the 10-min moving distance of the resin (1 g), which was placed on a glass slide with a slope of 60°. The TG-DTA analysis of the 3YSZ green body was performed in the temperature range of 30–1000 °C at a heating rate of 5 °C/min using DTG-60, SHIMADZU. The 3YSZ green body and sintered body were characterized by XRD analysis (max 2200 V, Rigaku Corporation, Tokyo, Japan) and field-emission scanning electron microscopy (FE-SEM, JSM-6700F, JEOL, Tokyo, Japan). For accurate microstructure analysis, the cross-section of the 3YSZ green body and sintered body was polished and thermal-etched at 1150 °C for 10 min before FE-SEM analysis. The relative density of the 3YSZ green body and sintered body was measured and calculated based on Archimedes’ method [15]. For the sintered body of the 3YSZ 3D-printed objects, the biaxial flexural strength was measured at a crosshead speed of 1 mm/min using a universal testing machine (AG-X plus, Kyoto, Japan), and the procedure and statistical analysis followed the ASTM standards ASTM C1161-18 and ASTM C1239.

## 3. Results and Discussion

For the supportless SLA 3D printing process, it is important to optimize the viscosity, photopolymerization properties, and flowability of UV-curable 3YSZ ceramic nanocomposite resins with 50 vol% of ceramic content. Therefore, these characteristics were investigated according to varying Bu (non-reactive diluent) content, as shown in Figure 1. Shear thinning behavior was observed in the viscosity analysis of all the 3YSZ nanocomposite resins, and the viscosity decreased with increasing Bu content, as shown in Figure 1a. All the 3YSZ nanocomposite resins had viscosities of 20,000–30,000 cps at a shear rate of 15.8 s^−1^, which is suitable for the supportless 3D printing process. As the shear rate was increased, the nanocomposite resins 3YSZ-Bu5% and 3YSZ-Bu10% showed stable behavior, but unstable behavior such as rapid decrease at a high shear rate was observed in 3YSZ-Bu0%. This property of 3YSZ-Bu0% may cause problems such as poor spreading characteristics during the 3D printing process. The photo-curing characteristics of the 3YSZ nanocomposite resins were examined at different Bu contents using photocalorimetry, as shown in Figure 1b. All the 3YSZ nanocomposite resins showed similar exothermic curves for 60 s, and the total photopolymerization amount increased with increasing Bu content. In other words, better photo-curing properties were observed as the Bu content increased, which appears to be a result of improvements in mobility and propagation ability of the generated radicals due to a good affinity between the monomer and Bu. Figure 1c shows the flowability of the 3YSZ nanocomposite resins, measured as the 10-min moving distance of the resin on a slide glass tilted at 60°. While moving distances of 2 mm or less are suitable for the flowability of the supportless 3D printing process, 3YSZ-Bu10% showed a moving distance of approximately 7 mm. However, 3YSZ-Bu0% and 3YSZ-Bu5% had moving distances of approximately 0.5 mm and 2 mm, respectively, which are suitable for the supportless 3D printing process. Considering these results, the 3YSZ nanocomposite with viscosity, photopolymerization property, and flowability suitable for high-content ceramic supportless 3D printing process was optimized

The green body of the 3YSZ 3D-printed objects had a high polymer content of 50 vol% compared to traditional ceramic samples for a sintering process, and the binding force between layers was weak because of the layer-by-layer additive method. As mentioned, layer separation and cracking can frequently occur due to the internal stress generated when materials such as solvents, residues, and polymers emerge from the objects during the drying and debinding processes (Figure 2). In order to minimize the internal stress generated from the removal of materials such as solvents and residues, the green body was studied under different drying conditions such as for 1 h at 80 °C in an oven and for 1 h at 25 °C under vacuum. The first weight losses of the green body during TGA were 2.04% and 0.68% before and after the drying process, respectively, which indicates that approximately 1.36% of the weight was removed through drying. Upon removal of this 1.36% weight, oven drying at 80 °C caused higher internal stresses because of a larger moving range and higher moving speed, requiring higher energy. Therefore, layer separation and cracking were frequently observed. In contrast, vacuum drying at 25 °C suppressed the formation of defects by gradually removing the material with low activity. To minimize the internal stress generated by removing the polymers with 50 vol% content, the dried green body was subjected to different debinding processes such as a two-step and a three-step process. For the 3YSZ ceramic nanoparticles, the manufacturer recommended a two-step holding time at 300 °C and 700 °C and a slow heating rate. However, because the TG-DTA analysis of the 3YSZ nanocomposite resin showed a sharp weight loss at 380 °C, a three-step debinding process with holding times at 300 °C, 380 °C, and 700 °C was performed. By providing an additional holding time at 380 °C where a rapid weight loss was observed, almost all the polymers could be slowly removed from the 3D-printed object. In other words, because almost no defects were generated in the 3YSZ 3D-printed object with a three-step debinding process, a high-density sintered body was obtained. From these results, the drying and debinding conditions for minimizing the internal stress in the 3YSZ 3D-printed objects were optimized to vacuum drying at 25 °C for 1 h and a three-step debinding process.

The XRD patterns of the green body and sintered body of 3YSZ 3D-printed objects were analyzed at different sintering temperatures for 2 h, as shown in Figure 3. Because the 3YSZ 3D-printed green body was a nanocomposite with a 3YSZ content of 50 vol%, it had a peak strength lower than those of the other 3YSZ 3D printing sintered bodies. In the XRD patterns of the 3YSZ 3D-printed green body, monoclinic peaks were observed at 2*θ* = 27–33°, and single peaks were observed at 2*θ* = 50–51° and 59–61°. On the contrary, the XRD results of the 3YSZ 3D-printed sintered body at different temperatures showed no monoclinic peaks at 27–33°, and a tetragonal peak was observed more clearly as the sintering temperature increased. Furthermore, tetragonal separated peaks for the (202) and (220) planes at 50–51° and those of (113) and (311) at 59–61° were more clearly observed as the sintering temperature increased. This is because at a higher sintering temperature more tetragonal phase is formed, and the disappearance of the monoclinic phase reduces. The optical images (inset) and SEM images of the green body and sintered body of 3YSZ 3D-printed objects were observed at different sintering temperatures, as shown in Figure 4. The optical image shows that the sintered body shrank in volume compared to the green body and was sintered very well without any defects such as cracks. Sintering is generally associated with remarkable shrinkage of the material because of the densification of the grain structure, thereby considerably reducing the porosity of the material. Hence, grain growth and grain boundaries are scarcely observed in the SEM image of the 3YSZ 3D-printed green body, and micropores are formed between the grains. In contrast, the grain size gradually increased in the 3YSZ 3D-printed sintered body as the sintering temperature increased, which is consistent with the XRD observation that the tetragonal peaks became clearer at higher sintering temperatures. Furthermore, the grain boundary was observed more clearly as the sintering temperature increased, but the grain boundary of the sintered body at the over-sintered temperature of 1500 °C was slightly blurred.

As the sintering temperature increased, although the shrinkage in diameter (*x* and *y* axis) was almost the same at approximately 14%, a minimum of 25% and a maximum of 30% shrinkage in the thickness (*z* axis, additive direction) were observed at 1150 °C and 1450 °C, respectively, as shown in Figure 5a. The volume shrinkage also increased from 45% to 49%. Unlike traditional ceramic sintering processes, thickness shrinkage seems to be the main determinant of volume shrinkage in the sintering process of 3D-printed ceramics. The relative density and grain size of the green body and sintered body of the 3YSZ 3D-printed objects were analyzed according to the sintering temperature, as shown in Figure 5b. A three-step sintering mechanism has been reported based on sintering theory [16]. For 3YSZ, the first stage is carried out at approximately 1150 °C. This stage, which is dominated by particle rearrangement such as sliding and rotation of grains, leads to the formation of grain boundaries and an increase in the grain size. In other words, in the first stage, the sintered body has a more compacted microstructure, and there only a few voids between the grains Therefore, the 3YSZ 3D-printed sintered body at 1150 °C had a higher relative density of 99.67% and larger grain size of 1.93 µm compared to the green body with 99.15% relative density and 1.19 µm grain size. The second stage, governed by cooperative particle rearrangement and atomic diffusion, occurs at approximately 1250 °C, when the open pores of the sintered body completely disappear. As the sintering temperature increased to 1450 °C, strong grain growth occurred, making the sintered body more compact, and the maximum values of the relative density of 99.91% and grain size of 2.64 μm were observed. In the 3YSZ 3D-printed objects sintered at 1550 °C above the third stage temperature, abnormal grain growth leads to a decrease in density due to pores trapped by the extreme sintering process, and a slightly lower relative density of 99.88% was observed. This is because the 3YSZ 3D-printed objects were sintered at an extreme sintering temperature, and the grain boundary of the sintered body became unclear, as shown in Figure 4. The flexural strengths of the green and sintered bodies of the 3YSZ 3D-printed objects were also observed according to the sintering temperature, as shown in Figure 5c. While the green body had a very low strength of 81.6 MPa, the flexural strength of the sintered body increased as the sintering temperature increased. The maximum flexural strength of 1002.5 MPa was observed for the objects sintered at 1450 °C, which is comparable to that of 1038 MPa for 3YSZ sintered by the traditional method [17]. Objects sintered at 1550 °C, which corresponds to an extreme sintering temperature, had a slightly lower flexural strength of 973.6 MPa owing to an unclear grain boundary. From these observations, the sintering temperature for the 3YSZ 3D-printed objects, in order to have high density and high strength, was optimized to 1450 °C.

The XRD patterns of the 3YSZ 3D-printed sintered body were analyzed at different sintering times at 1450 °C, as shown in Figure 6. As the sintering time increased, the tetragonal XRD peaks, such as those for (002) and (200) at 34–36°, (202), (220) at 50–51°, and (113) and (311) at 59–61°, were more clearly observed, and the crystal phase of 3YSZ changed from the monoclinic phase to the tetragonal phase. Furthermore, since the peak separation at 34–36°, 50–51°, and 59–61° was more clearly observed in the sintered body with sintering times of more than 60 min, it can be inferred that a sintering time of 60 min or more is required to sufficiently develop the tetragonal phase. SEM images of the 3YSZ 3D-printed sintered body were observed for different sintering times at 1450 °C, as shown in Figure 7. In all the 3YSZ 3D-printed sintered bodies sintered at 1450 °C, changes such as increasing grain size, reduced voids between the grains, and clearer grain boundaries were observed with an increase in sintering time; however, the grain boundaries of the body over-sintered for 180 min were slightly blurred.

The percentage of shrinkage of the 3YSZ sintered body was observed at different sintering times at 1450 °C, as shown in Figure 8a. As the sintering time increased, the volume shrinkage increased from 40% to 51%, and was mainly influenced by the percentage shrinkage in thickness, in the same manner as the effect of the sintering temperature. The relative density and grain size of the 3YSZ 3D-printed sintered body were analyzed according to the sintering time, as shown in Figure 8b. While the grain sizes of the sintered bodies with sintering times of 0 and 30 min were similar, the grain size also increased as the sintering time increased above 60 min. This is consistent with the XRD results showing that tetragonal peak separation became clearer in the sintered body when the sintering time was greater than 60 min. The relative density according to the sintering time gradually increased as the sintering time increased, but the maximum relative density of 99.95% was observed when sintering for approximately 150 min, which is similar to that for the conventional 3YSZ fine ceramic [18]. The relative density of the sintered body sintered for 180 min decreased because more than the allowable heat energy of the grain was supplied, and the grain boundary began to collapse. The flexural strength of the 3YSZ 3D-printed sintered body according to sintering time was observed, as shown in Figure 8c. While the flexural strength gradually increased as the sintering time increased, the maximum flexural strength of 1008.5 MPa was observed at a sintering time of 150 min. The flexural strength of the body sintered for 180 min decreased to 994.5 MPa because of a decrease in the relative density due to the collapse of the grain boundary. Therefore, to improve characteristics such as density and strength, the sintering conditions for the 3YSZ 3D-printed objects were optimized to a sintering temperature of 1450 °C and a sintering time of 150 min.

## 4. Conclusions

A 3YSZ 3D-printed sintered body with a high relative density of 99.95% and high flexural strength of 1008.5 MPa was manufactured by supportless SLA 3D printing technology. In order to be suitable for the supportless 3D printing process, the viscosity and flowability of the UV-curable 3YSZ ceramic nanocomposite resins with a ceramic content of 50 vol% were controlled to 20,000–30,000 cps and less than 2 mm moving distance, respectively. In order to prevent defect generation, the green body of the 3YSZ 3D-printed objects was dried in a vacuum oven for 24 h, followed by a three-step debinding process. The sintering temperature and sintering time for the sintered body were optimized to 1450 °C and 150 min, respectively, to achieve high strength. This is the start of a promising era for fabrication of ceramic 3D-printed objects with complex shapes, remarkable strengths, and unique ceramic functionality.

## Figures and Tables

**Figure 1 nanomaterials-11-00192-f001:**
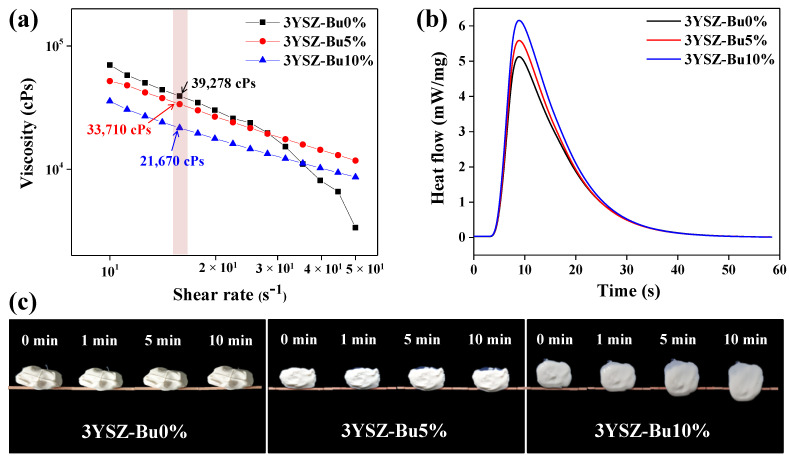
Characteristics of UV-curable 3YSZ ceramic nanocomposite resins with different Bu content: (**a**) Viscosity behavior, (**b**) heat flow curves, and (**c**) flowability.

**Figure 2 nanomaterials-11-00192-f002:**
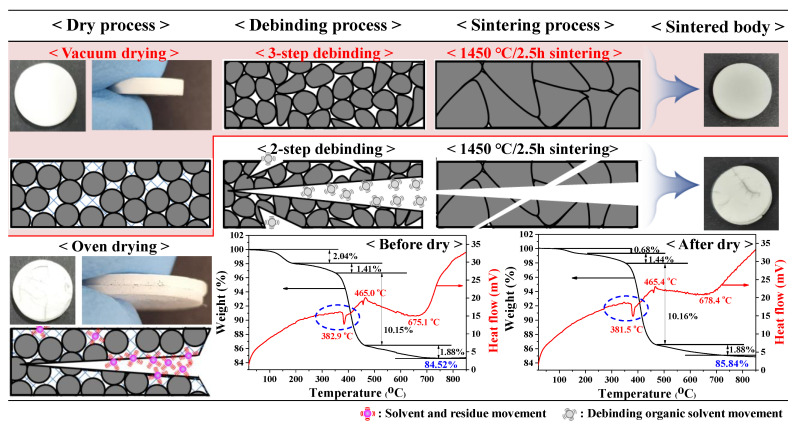
Schematic diagram for optimization of drying and debinding conditions for 3YSZ 3D-printed objects to minimize internal stress generated from removing materials such as solvents, residues, and polymers based on TG-DTA analysis.

**Figure 3 nanomaterials-11-00192-f003:**
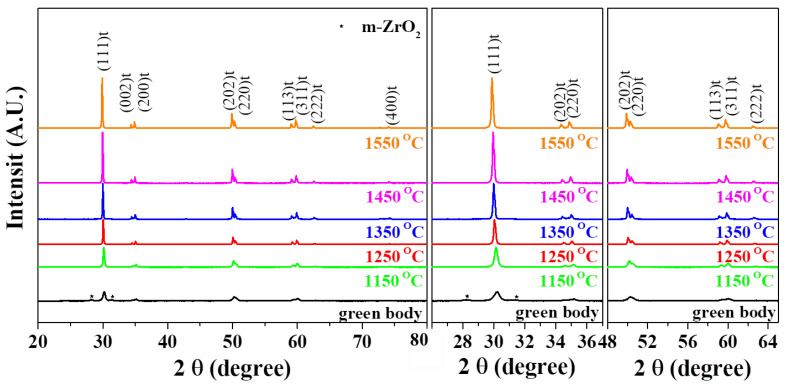
XRD patterns of 3YSZ 3D-printed sintered body according to sintering temperature.

**Figure 4 nanomaterials-11-00192-f004:**
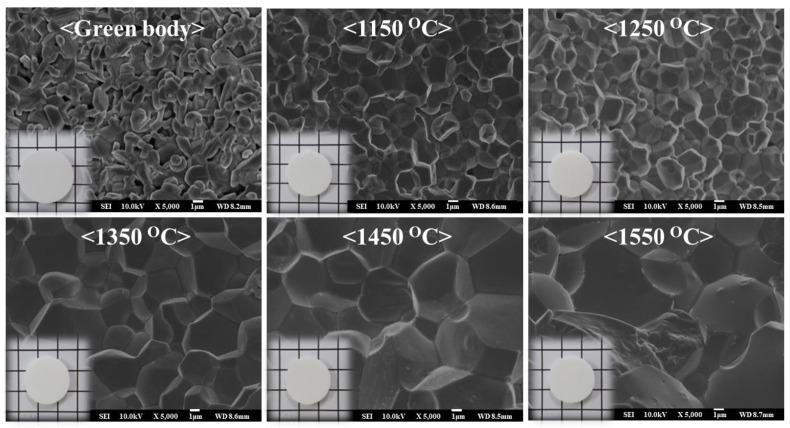
SEM images of 3YSZ 3D-printed sintered body according to sintering temperature. The inset shows optical images of the 3YSZ 3D-printed sintered body after the sintering process.

**Figure 5 nanomaterials-11-00192-f005:**
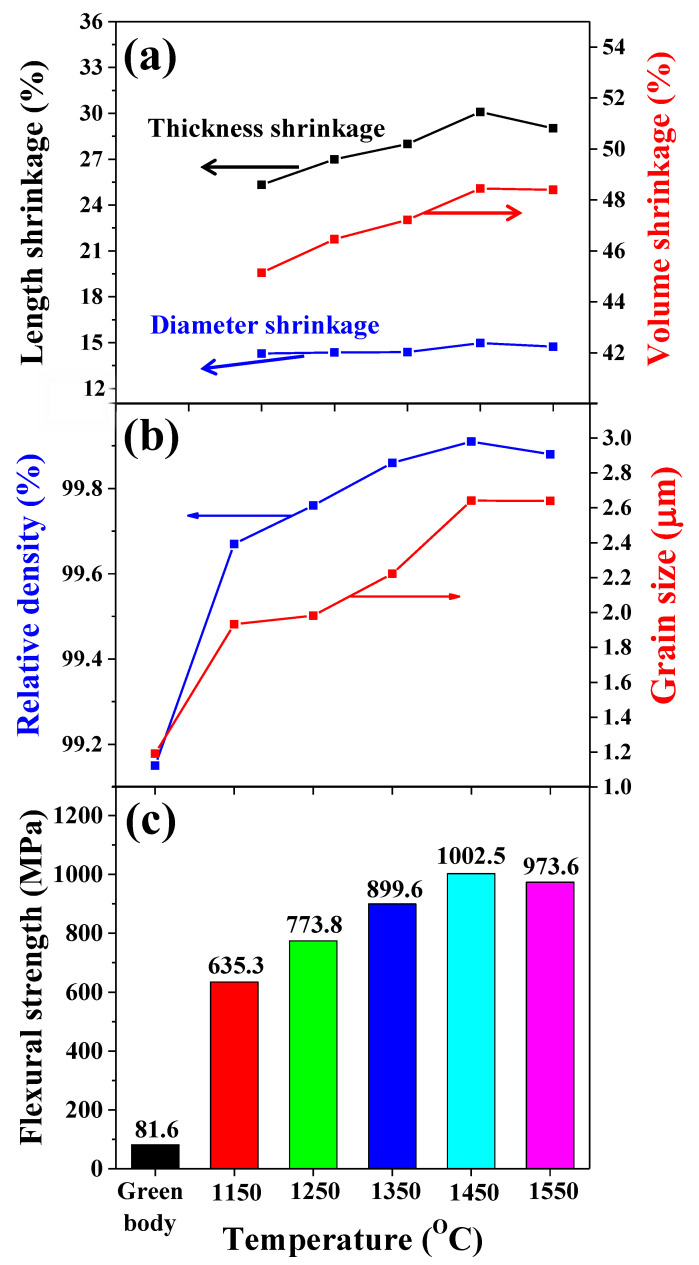
Characteristics of 3YSZ 3D-printed sintered body according to sintering temperature: (**a**) Length and volume shrinkage, (**b**) relative density and grain size, and (**c**) flexural strength.

**Figure 6 nanomaterials-11-00192-f006:**
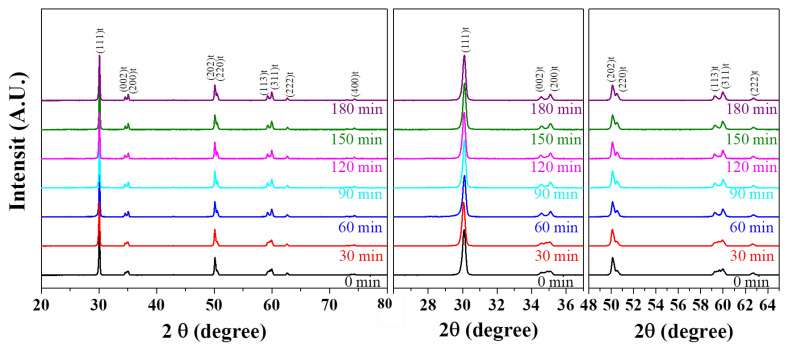
XRD patterns of 3YSZ 3D-printed sintered body according to sintering time.

**Figure 7 nanomaterials-11-00192-f007:**
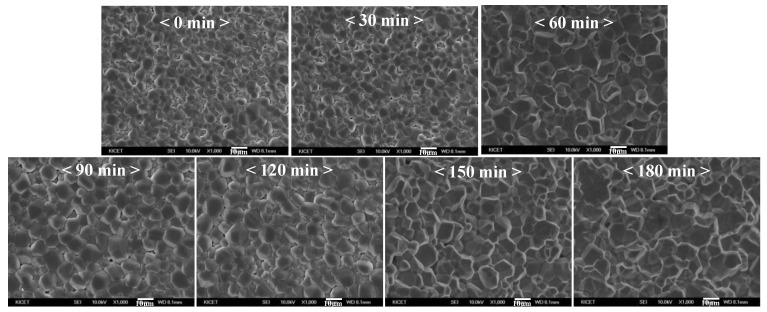
SEM images of 3YSZ 3D-printed sintered body according to sintering time.

**Figure 8 nanomaterials-11-00192-f008:**
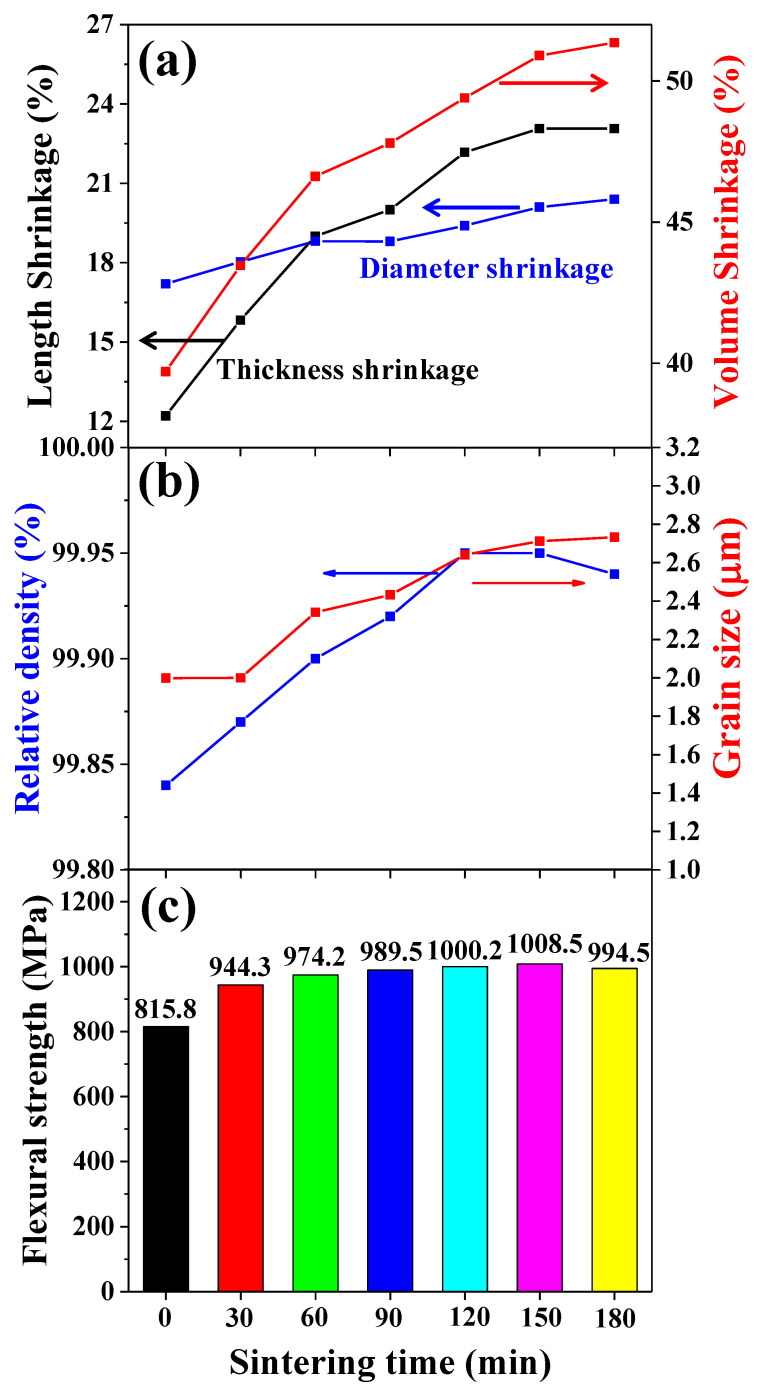
Characteristics of 3YSZ 3D-printed sintered body according to sintering time: (**a**) Length and volume shrinkage, (**b**) relative density and grain size, and (**c**) flexural strength.

**Table 1 nanomaterials-11-00192-t001:** Defined sample names of UV-curable 3YSZ (3 mol% yttria-stabilized zirconia) ceramic nanocomposite resins with different Bu content.

Sample Name	3YSZ (vol%)	HDDA (vol%)	Bu (vol% of Monomer)	BAPO (wt% of Monomer)	BYK111 (wt% of Particles)
**3YSZ-Bu0%**	50	50	0	2	2
**3YSZ-Bu5%**			5		
**3YSZ-Bu10%**			10		

## Data Availability

Data presented in this study are available by requesting from the corresponding author.

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
