# Peer review of "Sintering Process Optimization for 3YSZ Ceramic 3D-Printed Objects Manufactured by Stereolithography"

_nanomaterials, 2021, doi:10.3390/nano11010192_

Round 1
Reviewer 1 Report
The manuscript of the authors S. H. Ji et al. reports results of investigation where they developed drying, debinding and sintering regime for sintering 3D printed 3YSZ ceramic pellets.
It seems that the experimental work was well done, however the presentation and description of the results do not meet required criteria for publication.
- In general, sinterability of the particular particles depends on their nature, particle size and size distribution, morphology and other processing parameters. So, it is suggested, that the authors also include in the manuscript a micrograph of the starting powder. Even more, the authors repeated in the manuscript on many places that they started/deal with nanocomposites. In the Experimental part they mention that they used 3YSZ with particle size of 40 nm. However, in Fig. 3b on the picture of very porous microstructure of the green body there can be seen that the primarily units which are connected together, have a size from 1 to approximately 3 microns. This should be explained. In addition, on the SEM micrographs there should be larger magnification bar.
- The authors claim, that they used ZrO2 stabilized with 3% Y2O3 in order to avoid cracking. This should be explained. What was the aim of the investigation; To prepare monoclinic, tetragonal or cubic zirconia?
- The experimental part is well described however, the authors did not mention how they measure a density. Even more, the authors write density values very precisely with two decimal digits. I do not believe in such high precision.
- The authors studied microstructural properties of their pellets in detail, based on the surface microstructure of the as-prepared samples. This is my main concern about this manuscript. Surface properties may not be the same as bulk properties. For correct microstructural analysis, cross-section of samples should be polished and properly etched. In our laboratory we always use this approach for studying microstructure. I suggest that the journal editor decide whether all SEM micrographs should be replaced by the appropriate ones.
Reviewer 2 Report
The work has a typically technological character and contains almost no scientific elements. (Although the macroscopic analysis of the sintering process was made on a theoretical basis.) Only the demonstrated relationship between the mechanical properties of ceramics, density and microstructure contribute scientific elements. The published results of the thermal analysis indicate that a heated green body in the temperature range of 320-450 °C shows a rapid loss of weight with an accompanying exothermic effect - which may be decisive for obtaining a ceramics without microstructure defects and therefore the debinding step should be performed at these temperatures.The quality of these implementation studies should be assessed highly, as the 3D printing method may be useful for the practice of producing ceramics
Reviewer 3 Report
In the materials and methods section, it would be advisable to use a table to indicate the content of additives in each sample.
Figure 2 should be clarified, it is a bit confusing in the present form.
Figures 3 and 5 should be prepared in two figures each. On the one hand, indicate the XRD results and on the other hand the SEM results.
Round 2
Reviewer 1 Report
The authors have improved the manuscript and I believe that it meets required criteria for publication.